

# Recalculation of error growth models' parameters for the ECMWF forecast system

Hynek Bednář [1], Aleš Raidl [1] and Jiří Mikšovský [1]

[1]Department of Atmospheric Physics, Faculty of Mathematics and Physics, Charles University, Prague, 180 00, Czech Republic

*Correspondence to*: Hynek Bednář (hynek.bednar@mff.cuni.cz)

**Abstract.** This article provides a new estimate of error growth models' parameters approximating predictability curves and their differentials, calculated from data of the ECMWF forecast system over the 1986 to 2011 period. Estimates of the largest Lyapunov exponent are also provided, along with model error and the limit value of the predictability curve. The proposed correction is based on the ability of the Lorenz's (2005) system to simulate predictability curve of the ECMWF forecasting system and on comparing the parameters estimated for both these systems, as well as on comparison with the largest Lyapunov exponent ($\lambda = 0.35$ day$^{-1}$) and limit value of the predictability curve ($E_\infty = 8.2$) of the Lorenz's system. Parameters are calculated from the Quadratic model with and without model error, as well as by the Logarithmic and General models and by the hyperbolic tangent model. The average value of the largest Lyapunov exponent is estimated to be in the <0.32; 0.41> day$^{-1}$ range for the ECMWF forecasting system, limit values of the predictability curves are estimated with lower theoretically derived values and new approach of calculation of model error based on comparison of models is presented.

## 1. Introduction

Forecast errors in numerical weather prediction systems grow in time because of the inaccuracy of the initial state (initial error), chaotic nature of the weather system itself, and the model imperfections (model error). The growth of forecast error in weather prediction is exponential on average. As an error becomes larger, its growth slows down and then stops with the magnitude saturating at about the average distance between two states chosen randomly from dynamically and statistically possible states. This average growth of forecast error with increasing lead times is called the predictability curve.

Predictability curves (Froude et al., 2013) of the European Centre for Medium-Range Weather Forecasts (ECMWF) numerical weather prediction system are calculated by the approach developed by Lorenz (1982), where two types of error growth can be obtained (Lorenz, 1982). The first type is calculated as the root mean square difference between forecast data of increasing lead times and analysis data valid for the same time. This error growth estimate consists of initial and model error and following Lorenz (1982) we will call it the *lower bound predictability curve* (L). The second type is calculated as the root-mean-square difference between pairs of forecasts, valid for the same time but with times differing by some fixed time interval (the difference between two forecasts issued with 24-h lag but valid at the same time is used in this article). This type consists of



initial error and we will call it the *upper bound predictability curve* (U). Predictability curves of Lorenz's 05 system (L05; Lorenz, 2005) can be controlled by model parameters and by the size of the initial error and they are set to be as close to predictability curves of ECMWF forecasting system as possible.

Over the years several error growth models approximating predictability curves have been developed, aiming to quantify Lyapunov exponents, model errors (for the imperfect model case where the atmosphere is not perfectly modeled), and limit

(saturated) errors. The first, called Quadratic ($Km$), was designed by Lorenz (1969). Dalcher and Kalney (1987) added model error to the Quadratic model and Savijarvi (1995) changed it to the form ($Km_\beta$), that is used today. An alternative, called Logarithmic model ($Lm$) was introduced by Trevisan et al. (1992; 1993). General model ($Gm$) was introduced by Stroe and Royer (1993; 1994). All these models approximate differences of predictability curves (error growth rate). Newer models approximate the predictability curve directly by the hyperbolic tangent ($Tm$ and $Tm_\beta$) (Žagar et al., 2017).

Values of parameters calculated from error growth models are used to evaluate the improvement of the ECMWF forecasting system (Magnusson and Kallen, 2013), to estimate the predictability or the limit error (Bengtsson et al., 2008), to quantify impacts of different model's resolutions (Buizza, 2010), to study chaos and model error in different spatial-temporal scales (Žagar et al., 2015; Žagar et al., 2017 ). They are also used by researchers when the need arises to estimate chaoticity, model error or predictability, but their validity can't be proved, because standard methods (Sprott, 2006) to calculate the largest

Lyapunov exponents for the ECMWF forecasting system can't be used due to a large number of variables. An independent value estimated from forecast and analysis anomalies can be calculated for the limit error (Simmons et al., 1995) and its validity will be discussed.  The need for correct values of error growth models´ parameters increased these days because the Quadratic model with model error is used to describe multiscale weather (Zhang et al., 2019).

This article intends to provide a new estimate of parameters of error growth models in the ECMWF forecasting system

calculated from data over the 1986 to 2011 period. The correction is based on comparing the parameters calculated from the error growth models for the L05 system and the ECMWF forecasting system and on comparison with the largest Lyapunov exponent and the limit value of the predictability curve of the L05 system that can be calculated independently and with sufficient accuracy. To make the correction valid, predictability curves of the ECMWF forecasting system and the L05 systems are compared for two different methods (arithmetic and geometric averages) and the number of variables of the L05 system,

pertaining to the best match of the predictability curves is identified. As a result, a new approach to the calculation of model error based on a comparison of models is presented.

This article is divided into six sections.  The second describes the experimental setting. The third provides a comparison of predictability curves of the ECMWF forecasting system and the L05 system and the fourth deals with the estimation of Lyapunov exponents, model, and limit errors of the ECMWF forecasting system based on the correction.  Discussion and

conclusions are then presented in the final two sections.



## 2. Experimental setting

L05 model is based on the low-dimensional atmospheric system presented by Lorenz (1996). It is a nonlinear model, with N variables connected by governing equations

$$dX_n/dt = -X_{n-2}X_{n-1} + X_{n+1}X_{n-1} - X_n + F, \qquad (1)$$

$n = 1,...,N$ . $X_{n-2}$ , $X_{n-1}$, $X_n$ , $X_{n+1}$ are unspecified (i.e., unrelated to actual physical variables) scalar meteorological quantities, F is a constant representing external forcing and t is time. The index is cyclic so that $X_{n-N} = X_{n+N} = X_n$ and variables can be viewed as existing around a circle. Nonlinear terms of Eq. (1) simulate advection. Linear terms represent mechanical and thermal dissipation. The model quantitatively, to a certain extent, describes weather systems, but, unlike the well-known Lorenz's model of atmospheric convection (Lorenz, 1963), it cannot be derived from any atmospheric dynamic

equations. The motivation was to formulate the simplest possible set of dissipative chaotically behaving differential equations that share some properties with the "real" atmosphere. One of the model´s properties is to have 5 to 7 main highs and lows that correspond to planetary waves (Rossby waves) and a number of smaller waves that correspond to synoptic-scale waves. For Eq. (1) this is only valid for $N = 30$ and that is, as it will be seen, not sufficient for the experimental setting. Therefore, spatial continuity modification of L05 system is used, where the Eq. (1) is rewritten to the form:

$$dX_n/dt = [X,X]_{L,n} - X_n + F, \qquad (2)$$

where

$$[X,X]_{L,n} = \sum_{j=-J}^{J}{}' \sum_{i=-J}^{J}{}' \left( -X_{n-2L-i}X_{n-L-j} + X_{n-L+j-i}X_{n+L+j} \right) \Big/ L^2.$$

If $L$ is even, $\sum$' denotes a modified summation, in which the first and last terms are to be divided by 2. If $L$ is odd, $\sum$' denotes an ordinary summation. Generally, $L$ is much smaller than $N$ and $J = L/2$ if $K$ is even and $J = (L-1)/2$ if $L$ is odd. For comparison

with predictability curves of the ECMWF forecasting system, we choose $N$ =30; 60; 90; 120; 150; 360. To keep a desirable number of main pressure highs and lows, Lorenz (2005) suggested to keep ratio $N/L = 30$ and therefore $L = 1; 2; 3; 4; 5; 12$. For even values of $L$ we have $J = 1; 2; 6$ and for odd values of $L$ we have $J = 0; 1; 2$. Parameter $F = 15$ is selected as a compromise between too high Lyapunov exponent (smaller $F$) and undesirable shorter waves (larger $F$). For this setting and by the method of numerical calculation (Sprott, 2006), the global largest Lyapunov exponents are calculated (Table 2). By the

definition of Lorenz (1969): „A bounded dynamical system with a positive Lyapunov exponent is chaotic ". Because the value of the largest Lyapunov exponent is positive and the system under study is bounded, it is chaotic. Strictly speaking (Aligood





et al., 1996), we also need to exclude the asymptotically periodic behavior, but such a task is impossible to fulfill for the numerical simulation. The choice of parameters $F$ and time unit = 5 days is made to obtain a similar value of the largest Lyapunov exponent as the ECMWF forecasting system.

To calculate predictability curves (Lorenz, 1996), arbitrary values of the variables $X_n$ are chosen, and, using a fourth-order Runge-Kutta method with a time step $\Delta t = 0.05$ or 6 hours, it is integrated forward for 14400 steps or 10 years. Final values $X_{0,n}$, which should be free of transient effect, are the initial values of "reality". Initial values of "prediction" are $X'_{0,n} = X_{0,n} + e_{0,n}$, where $e_{0,n}$ is the initial error and it is chosen randomly from a normal distribution $ND(\mu;\sigma)$, where $\mu = 0$ is mean and $\sigma$ is the standard deviation, which is chosen from comparison with the ECMWF forecasting system. From $X_{0,n}$

and $X'_{0,n}$ Eqs. (2) are integrated forward for 37.5 days ($K$=150 steps). For upper bound predictability curves $X_n$ and $X'_n$ are chosen with the same number of variables $N$. For lower bound predictability curves $X_n$ is defined by $X_{0,n}$ and by Eqs. (2) with $N_0 = 360$ and $X'_n$ by $X'_{0,n}$ and by Eqs. (2) with $N = 30; 60; 90; 120; 150$. The size of the model error is corrected by the difference of $N$ for $X_n$ and $X'_n$. If, for example, $N = 120$ then $X_n$ is compared with $X'_n$ in each third point of $N_0$.

In each time step $\Delta t$ of numerical integration $N$ "real" and $N$ "observed" values are obtained. The size of the error at a given

time for upper bound predictability curves is $e_n(k \cdot \Delta t) = X'_{k,n} - X_{k,n}$, where $k = 1, \ldots, K$ and $n = 1, \ldots, N$ and for lower bound predictability curves $\varepsilon_n(k \cdot \Delta t) = X'_{k,n} - X_{k,n'}$, where $k = 1, \ldots, K$, $n = 1, \ldots, N$ (except for $N_0$). $n' = 1, \ldots, N$ (except for $N_0$) is the location of the value $X_{k,n'}$ for $N = 360$, where $n' = n \cdot N_0/N$ for $N = 30; 60; 90; 120; 150$. The predictability curves of the ECMWF forecasting system, in this case, are obtained from annual averages of daily data. To simulate that, the number of runs $M = 400$ is made. In each new run, initial values $X_{0,n}$ are the last values $X_{K,n}$ from the previous run. $M \cdot N$ values are

obtained for each $k$. Final formulas of prediction errors that constitute predictability curves by calculation with arithmetic mean (A) are:

$$E_{U(A)}^{L05}(k \cdot \Delta t) = \sqrt{\frac{1}{M \cdot N} \sum_{m=1}^{M} \sum_{n=1}^{N} e_{n,m}^2(k \cdot \Delta t)}, \tag{3}$$

$$E_{L(A)}^{L05}(k \cdot \Delta t) = \sqrt{\frac{1}{M \cdot N} \sum_{m=1}^{M} \sum_{n=1}^{N} \varepsilon_{n,m}^2(k \cdot \Delta t)}. \tag{4}$$

Formulas to calculate prediction errors by geometric means (G) are:

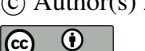



$$E_{U(G)}^{L05}\left(k\cdot\Delta t\right)= \sqrt[2M]{\prod_{m=1}^{M}\left(\frac{1}{N}\sum_{n=1}^{N}e_{n,m}^{2}\left(k\cdot\Delta t\right)\right)}, \tag{5}$$

$$E_{L(G)}^{L05}\left(k\cdot\Delta t\right)= \sqrt[2M]{\prod_{m=1}^{M}\left(\frac{1}{N}\sum_{n=1}^{N}\varepsilon_{n,m}^{2}\left(k\cdot\Delta t\right)\right)}. \tag{6}$$

For an overview of the symbols see Table 1.

To calculate predictability curves for the ECMWF forecasting system (EFS) values of 500 hPa geopotential height are used. Data were obtained from ECMWF (Magnusson, 2018). Lower bound predictability curves are calculated (Magnusson and Kallen, 2013) from twenty-one root mean squares over the Northern Hemisphere (20°–90° N) obtained daily from 1 January 1986 to 31 December 2011. Means are differences between operational forecasts and analyses from ERA-Interim for a given day. Forecasts range from 0.5 day ago relative to the given day to 10 days ahead, with time step 0.5 day. The difference between operational analysis and analysis from ERA-Interim is taken as the initial error. Upper bound predictability curves are calculated (Magnusson and Kallen, 2013) from twenty-seven root mean squares over Northern Hemisphere (20°–90°) obtained daily from 1 January 1986 to 31 December 2011. Means are differences between two operational forecasts issued with one day lag, but valid at the same day. Specifically, following differences are obtained for a given day (hours): 0–24, 6–30, 12–36, 18–42, 24–48, 30–54, 36–60, 42–66, 48-72, 54–78, 60–84, 66–90, 72–96, 78–102, 84–108, 90–114, 96–120, 108–132, 120–144, 132–156, 144–168, 156–180, 168–192, 180–204, 192–216, 204–228, 216–240. Prediction errors constituting the predictability curves are calculated as annual averages of daily data. Detailed information about calculating predictability curves of the ECMWF forecasting system can be found in Lorenz (1982).

Comparisons of model predictability curves are done through values normalized by the limit (saturated) errors ( $E_{\infty,U}=\lim_{t\to\infty}E_{U}$ , $E_{\infty,L}=\lim_{t\to\infty}E_{L}$ ). Because maximum forecast time for the ECMWF forecasting system is 10 days, presented predictability curves don't reach their limit value. Independent measure of limit error can be calculated as:

$$E_{\infty,L}=\sqrt{\overline{\left(f-c\right)}^{2}+\overline{\left(a-c\right)}^{2}}; \ \ E_{\infty,U}=\sqrt{2\overline{\left(f-c\right)}^{2}}, \tag{7}$$

where $\overline{\left(f-c\right)}$ is the time-averaged anomaly with respect to climate and $\overline{\left(a-c\right)}$ is the time-averaged analysis anomaly with respect to climate. The climate is defined from ERA-Interim daily climatology. More information can be found in (Simmons et al., 1995). Because it will be shown that values of limit error calculated by this method aren´t correct, predictability curves of the ECMWF forecasting system are normalized by values calculated by Eq. (15).





## 3. Comparison of predictability curves

Predictability curves of the ECMWF and L05 systems are compared to find a setting of the L05 system (number of variables ($N$), the size of the initial errors, preference of arithmetic or geometric mean) that gives the most similar progress of systems' predictability curves.

Predictability curves of the L05 system show negative growth for the first time step (6 hours) but turn into an increase thereafter. At the second time step (12 hours) values of predictability curves reach approximately the same values as it had initially. A possible explanation could be that initial errors set the initial state off the attractor and decrease occurs because the first tendency is to get on the attractor (Brisch and Kantz, 2019). With an increase of average errors, chaotic behavior becomes dominant. Predictability curves of the ECMWF forecasting system do not exhibit this type of behavior. This may be because of larger time steps or methods of objective analysis. We aim to get the most similar predictability curves of both models and therefore the first two time steps (up to 12 hours) of L05 model's predictability curves are filtered out.

Initials values $E_U^{L05}(0)$ and $E_L^{L05}(0)$ or equivalently standard deviations $\sigma$ from a normal distribution $ND(\mu;\sigma)$ of the L05 system are calculated from a comparison of values that are normalized ($E_{Norm}$) by limit (saturated) errors $E_\infty$ calculated by Eq. (15). Upper bound predictability curves start for the ECMWF forecasting system at day one (the difference between one-day prediction and the analysis) and therefore $E_U^{L05}(0)$ are calculated from predictability curves that are close at the first day $\left(E_{Norm}^{L05}(1)=E_{Norm}^{EFS}(1)\right)$. Values for the L05 system are computed for $N = 60; 90; 120; 150$. Normalized predictability curves with $N = 30$ exhibit different evolution compared to predictability curves of the ECMWF forecasting system and they aren't displayed. Initial prediction errors $E_U^{L05}(0)$ calculated by arithmetic and geometric mean and $N = 60; 90; 120; 150$ have the same values and $E_U^{L05}(0)\in\langle 0.3;0.8\rangle$. For lower bound predictability curves of the ECMWF forecasting system, the initial error $E_L^{EFS}(0)$ is computed as a difference between analysis from the operational forecasting system and analysis from ERA-Interim. Initial errors of the L05 system $E_L^{L05}(0)$ are calculated as: $E_L^{L05}(0)=E_{\infty,L}^{L05}\cdot E_L^{EFS}(0)\big/E_{\infty,L}^{EFS}$ and $E_L^{L05}(0)\in\langle 0.2;0.7\rangle$. Values are the same for all $N$ and arithmetic and geometric mean.

Predictability curves calculated by arithmetic and geometric mean show a significant difference for the L05 system (all $N$) and minor difference for the ECMWF forecasting system. For the L05 system and upper and lower bound predictability curves, the maximal difference is between 6.5 % and 10.5 % of $E_{\infty,U}^{L05}$ or $E_{\infty,L}^{L05}$ and these maximal values occur between 5 and 9 day of forecast length. For the ECMWF forecasting system and upper and lower bound predictability curves, the maximal difference is 2 % of $E_{\infty,U}^{EFS}$ or $E_{\infty,L}^{EFS}$ and these maximal values occur at the end of the forecast length (10 day). The choice of the averaging method doesn't significantly change the evolution of the ECMWF forecasting system's predictability curves and it does not change values of parameters of the approximations. For the L05 system, the choice of averaging method is significant and it changes values of the parameters. The reason for this sensitivity can be found in the spread of values that are



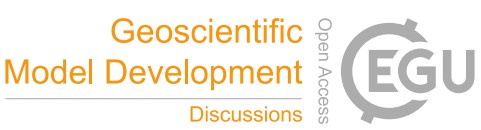

used for averaging. For the ECMWF forecasting system, the values are closer to each other than for the L05 system and from the definition of means, it leads to the aforementioned difference.

The comparison of predictability curves is done with given initial values. Predictability curves of the ECMWF forecasting system are normalized by $E_{\infty,U}^{EFS}$ or $E_{\infty,L}^{EFS}$ (Fig. 7, black full curves) and for the L05 system by $E_{\infty,U}^{L05}$ and $E_{\infty,L}^{L05}$ displayed in Table 2 (for a description of the symbols see Table 1). For the L05 system predictability curves are calculated with $N = 60$;

90; 120; 150 variables and by arithmetic and geometric mean. For the ECMWF forecasting system only arithmetic mean is used.

A comparison of lower bound predictability curves (Fig. 2) shows the most similar predictability curves of the ECMWF forecasting system and the L05 system for the L05 system calculated by arithmetic mean with $N = 90$. For upper bound predictability curves (Fig. 1), predictability curves for the L05 system with $N = 90$ are the most similar but to the year 1999

for predictability curves of the L05 system calculated by geometric mean and after 1999 by the arithmetic mean.

## 4. Estimation of parameters

Parameters of error growth models are the Lyapunov exponent, model error, and limit error. They are estimated from approximations of predictability curves or differences of predictability curves $\left(\left(E\left(t+\Delta t\right)+E\left(t\right)\right)/2;\left(E\left(t+\Delta t\right)-E\left(t\right)\right)/\Delta t\right)$, where $t$ is time and $\Delta t = 0.25\,\text{day}$ (Figs. 3 and 4). Error growth models considered here are:


$$Km := \frac{dE\left(t\right)}{dt} = \alpha E\left(1 - \frac{E}{E_{\text{lim}}}\right), \tag{8}$$

$$Km_{\beta} := \frac{dE\left(t\right)}{dt} = \left(\alpha E + \beta\right)\left(1 - \frac{E}{E_{\text{lim}}}\right), \tag{9}$$

$$Lm := \frac{dE\left(t\right)}{dt} = -\alpha E \ln\left(\frac{E}{E_{\text{lim}}}\right), \tag{10}$$

$$Gm := \frac{dE\left(t\right)}{dt} = \frac{\alpha}{p} E\left(1 - \left(\frac{E}{E_{\text{lim}}}\right)^{p}\right), \tag{11}$$

$$Tm := E\left(t\right) = A\tanh\left(at + a\right) + A, \tag{12}$$





where parameters of $Tm$ are $\alpha = 2a$, $E_{\text{lim}} = 2A$ and

$$Tm_\beta := E(t) = A\tanh(at+b)+B, \tag{13}$$

where parameters of $Tm_\beta$ are $\alpha = a(A+B)/A$, $\beta = a(A^2 - B^2)/A$ and $E_{\text{lim}} = A+B$. $E$ is an average forecast error. $t$ represents time, $\alpha$ is the estimate of the Lyapunov exponent $\lambda$. $\beta$ is the parameter of model error ($dE/dt$ when $E=0$), $E_{\text{lim}}$ is the limit (saturated) value of $E$ (value of $E$ when $dE/dt=0$, theoretically $E_\infty$) and $p$, $A$, $B$, $a$, $b$ are parameters.

The calculation is done for the ECMWF forecasting system and the L05 system ($N=90$), for arithmetic ($A$) and geometric ($G$) means, for upper bound predictability curves ($U$) and lower bound predictability curves ($L$). See Tables 3 and 4 for RMS values of parameters $\bar{\alpha}$, $\bar{E}_{\text{lim}}$, $\bar{\beta}$ and $\bar{p}$, that are calculated over all used initial errors for the L05 system and all calculated years for the ECMWF forecasting system.

The average values of parameters $\bar{\alpha}$, $\bar{E}_{\text{lim}}$ are higher for the lower bound predictability curves than for the upper bound

predictability curves. Upper bound predictability curves should not include model error (theoretically $\beta=0$) but from Table 4 it can be seen that for the L05 system (arithmetic mean) the values are even higher than for the lower bound predictability curves. For the ECMWF forecasting system the values of $\bar{\beta}$ are higher for lower bound predictability curves which is theoretically more acceptable, but $\bar{\beta}$ is not zero for the upper bound predictability curves. A possible explanation can be the sensitivity to correct approximation (cases with higher $\beta$ have lower $\alpha$), but this can not fully explain the discrepancy. For

$\bar{p}$ the values of upper and lower bound predictability curves are similar to each other (L05 system and ECMWF forecasting system).

There are significant differences of parameters $\bar{\alpha}$, $\bar{E}_{\text{lim}}$, $\bar{\beta}$ and $\bar{p}$ between predictability curves calculated by arithmetic and geometric mean for the L05 system (for the ECMWF forecasting system only arithmetic mean is presented). The most significant differences are detected for $\bar{\beta}$ and $\bar{p}$, where for $\bar{\beta}$ values are closer to zero for geometric mean and values of

predictability curves calculated by arithmetic mean are two or three times higher. Values of parameter $\bar{p}$ are closer to $\bar{p}=1$ for geometric mean. This means that differences of predictability curves calculated by geometric mean have a shape that is close to a symmetric parabola (for example Fig. 3a) but for the arithmetic mean the parabolic shape is skewed to the left (for example Fig. 3c).

The Lyapunov exponent of the ECMWF forecasting system is recalculated by the formula:

$$\lambda^{EFS} = \alpha^{EFS} + (\lambda^{L05} - \alpha^{L05}), \tag{14}$$





where $\alpha^{EFS}$ and $\alpha^{L05}$ are parameters of error growth models and $\lambda^{L05} = 0.35$ day$^{-1}$. For upper bound predictability curves (the L05 system with $N = 90$, to the year 1999 calculated by geometric mean and after 1999 by arithmetic mean), the average value $\bar{\lambda}_U^{EFS}$ over all error growth models is in the range $\langle 0.33;\ 0.41 \rangle$ day$^{-1}$ (Fig. 5a). *Lm* is not used, because this error growth model is not sufficient to approximate predictability curves. RMSEs of $\bar{\lambda}_U^{EFS}$ are mostly about 0.01 day$^{-1}$ only in years 1991,

1995, 1997 a 1999 RMSE is about 0.02 day$^{-1}$. For comparison, RMSEs of $\bar{\alpha}_U^{EFS}$ are in the range $\langle 0.02;\ 0.07 \rangle$ day$^{-1}$ (Fig. 5a). For lower bound predictability curves (the L05 system with $N = 90$ calculated by arithmetic mean), the average value $\bar{\lambda}_U^{EFS}$ over all error growth models is in the range $\langle 0.32;\ 0.41 \rangle$ day$^{-1}$ (Fig. 5b). RMSEs of $\bar{\lambda}_L^{EFS}$ are in the range $\langle 0.01;\ 0.02 \rangle$ day$^{-1}$. For comparison, RMSEs of $\bar{\alpha}_L^{EFS}$ are in the range $\langle 0.03;\ 0.07 \rangle$ day$^{-1}$ (Fig. 5b). The average value $\bar{\bar{\lambda}}^{EFS}$ over upper and lower bound predictability curves is shown in Fig. 6 and RMSEs of $\bar{\bar{\lambda}}^{EFS}$ are mostly about 0.01 d$^{-1}$. Low values of RMSEs of $\bar{\lambda}^{EFS}$

compared to RMSEs of $\bar{\alpha}^{EFS}$ and similar values of $\bar{\lambda}^{EFS}$ for upper and lower bound predictability curves (low values of RMSEs of $\bar{\bar{\lambda}}^{EFS}$) prove the validity of $\bar{\bar{\lambda}}^{EFS}$. Values of $\bar{\bar{\lambda}}^{EFS}$ and $\bar{\lambda}^{EFS}$ are generally closer to parameters $\alpha^{EFS}$ of $Km_\beta$, $Tm_\beta$ and $Gm$ than to $\alpha^{EFS}$ of $Km$, $Tm$ and $Lm$, but none of the error growth models approximates $\bar{\bar{\lambda}}^{EFS}$ (Fig. 6).

New limit values $E_\infty^{EFS}$ are calculated from the error growth models by the formula:

$$E_\infty^{EFS} = E_{\lim}^{EFS} + \left( E_{\lim}^{EFS} \cdot \left( E_\infty^{L05} - E_{\lim}^{L05} \right) \right) \Big/ E_\infty^{L05}, \tag{15}$$

where $E_\infty^{L05}$ and $E_{\lim}^{L05}$ are values from error growth models and $E_\infty^{L05} = 8.2$. For upper bound predictability curves (the L05 system with $N = 90$), average value over all error growth models $\bar{E}_{\infty,U}^{EFS}$ is in the range $\langle 96;\ 133 \rangle$ m (Fig. 7a). *Lm* is not used, because this error growth model is not sufficient to approximate predictability curves. RMSEs of $\bar{E}_{\infty,U}^{EFS}$ are mostly about 1 m only in the years 1987, 1988, 1995, 1997, 2003, and 2011 it is about 2 m. For comparison, RMSEs of $\bar{E}_{\lim,U}^{EFS}$ are in the range $\langle 2;\ 6 \rangle$ m (Fig. 7a). For lower bound predictability curves (the L05 system with $N = 90$ calculated by arithmetic mean), average

value over all error growth models $\bar{E}_{\infty,L}^{EFS}$ is in the range $\langle 114;\ 134 \rangle$ m (Fig. 7b). *Lm* is not used, because this error growth model is not sufficient to approximate predictability curves. RMSEs of $\bar{E}_{\infty,L}^{EFS}$ are mostly 3 m and after the year 2004, they are 4 m. RMSEs of $\bar{E}_{\lim,L}^{EFS}$ are in the range $\langle 3;\ 6 \rangle$ m (Fig. 7b). Lower values of RMSEs of $\bar{E}_{\infty,U}^{EFS}$ and $\bar{E}_{\infty,L}^{EFS}$ calculated by Eq. (15) compared to RMSEs of $\bar{E}_{\lim,U}^{EFS}$ and $\bar{E}_{\lim,L}^{EFS}$ prove the validity of $\bar{E}_{\infty,U}^{EFS}$ and $\bar{E}_{\infty,L}^{EFS}$.

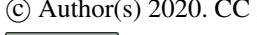



## 5. Discussion

The argument that favors $E_\infty^{EFS}$ calculated by Eq. (15) (Fig. 7, black full curves) instead of $E_\infty^{EFS}$ calculated by Eq. (7) (Fig. 7,

black dashed curves) is based on the parameter of model error $\beta$. The most similar predictability curves of the L05 system

and the ECMWF forecasting system with $E_\infty^{EFS}$ calculated by Eq. (15) are found for the L05 system with $N = 90$ (for lower

bound predictability curves calculated by arithmetic mean and for upper bound predictability curves calculated by geometric

mean to 1999 and after by arithmetic mean). The most similar predictability curves of the L05 system and the ECMWF

forecasting system with $E_\infty^{EFS}$ calculated by Eq. (7) are found for the L05 system with $N = 90$ by the arithmetic mean for upper

and lower bound predictability curves. It means that if the comparison is valid and model error is constant for the L05 system

(same number of variables over years in the L05 system means constant model error over years), it must be constant also for

the ECMWF forecasting system, but the calculation of parameters $\beta_L^{EFS}$ shows a decreasing trend with increasing time (Fig.

8b). This can't help yet. But parameters $\beta_U^{EFS}$ have non zero values (Fig. 8a) that are close to $\beta_L^{EFS}$ for some years and that

is inconsistent with the theoretical expectation that upper bound predictability curves should be without model error and

therefore $\beta$ should be 0 m/day. This inconsistency can be solved by the new definition of the model error. From Fig. 6 it can

be seen closer value of $\alpha^{EFS}$ to $\bar{\bar{\lambda}}^{EFS}$ for $\alpha^{EFS}$ approximated from error growth models $Km_\beta$ , $Tm_\beta$ and $Gm$ than for $\alpha^{EFS}$

approximated from error growth models $Km$, $Tm$ and $Lm$. $Gm$ has parameter $p$ that defines skewness of the originally

parabolic shape of the difference of predictability curves. $p = 1$ pertains to symmetrical parabolic shape ( $Gm$ becomes $Km$ )

and $p = 0$ means the greatest skewness to the left ( $Gm$ becomes $Lm$ ). Parameters $\beta$ also skew the originally parabolic shape

(Figs. 3 and 4). The model error can be seen as a difference between skewness of upper and lower bound predictability curves

and the new definition of model error would be:

$$\beta_{L-U} = \left| \beta_L - \beta_U \right|. \tag{16}$$

   Results (Fig. 9a) show good agreement for $\beta_{L-U}^{EFS}$ (Eq. (16)) calculated from $Km_\beta$ and $Tm_\beta$ , decreasing trend of $\beta_{L-U}^{EFS}$ with

increasing time for predictability curves with $E_\infty^{EFS}$ calculated by Eq. (15) and almost constant values of $\beta_{L-U}^{EFS}$ with increasing

years (slight decrease can be due to the error of approximations) for predictability curves with $E_\infty^{EFS}$ calculated by Eq. (7).

There is also good agreement with trends of $\left| p_L - p_U \right|$ (Fig. 9b). Because constant values of $\beta_{L-U}$ for predictability curves

with $E_\infty^{EFS}$ calculated by Eq. (7) are not theoretically possible, predictability curves with $E_\infty^{EFS}$ calculated by Eq. (15) are

favored. The reason for the decreasing trend of $\beta_{L-U}^{L05}$ , found for predictability curves of the L05 system with $N = 90$ that are



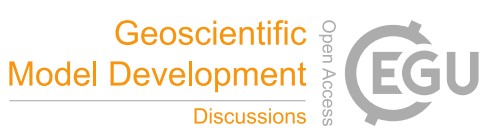

the most similar with predictability curves of the ECMWF forecasting system normalized by $E_\infty^{EFS}$ calculated by Eq. (15), is

that they are partly calculated by geometric and partly by the arithmetic mean.

These arguments are taken as proof of the validity of $\bar{\bar{\lambda}}^{EFS}$, $E_\infty^{EFS}$ calculated by Eq. (15). The reason for the overestimation

of $E_\infty^{EFS}$ calculated by Eq. (7) (Fig. 7) can be found in the multiscale behavior of weather. If some events are predictable on a

timescale longer than ten days (for example long-lived anomalies in sea surface temperature or soil moisture) than they

wouldn´t be captured by medium-range weather forecast (Simmons et al., 1995; Brisch and Kantz, 2019). It is also possible

that the overestimation is due to the different source of data used for calculation of $E_\infty^{EFS}$ by Eqs. (7) and (15): For $E_\infty^{EFS}$

calculated by Eq. (7) only data from ERA-Interim (Janoušek 2011) are used but for $E_\infty^{EFS}$ calculated by Eq. (15) data from

operational forecast are employed.

At the end of this section, it is important to remind the readers about the importance of the correct values of the parameters.

Nowadays, $Km_\beta$ is used in the ECMWF forecasting system to estimate the influence of different spatiotemporal scales where

parameter $\beta$ newly represents the intrinsic upscale error growth and propagation from small scales and $\alpha$ represents synoptic-

scale error growth (Zhang et al., 2019). The results of our analysis well support this approach by the new definition of model

error (Eq. (16)) and by showing the errors of approximations for individual error growth models.

## 6. Conclusion

The values of error growth models' (Eqs. (8) - (13)) parameters that approximate predictability curves and their differences

(Figs. 3 and 4) in the ECMWF forecast system (Tables 3 and 4) were recalculated. It is based on similarities of normalized

upper and lower bound predictability curves (Figs. 1 and 2) of the ECMWF forecasting system (annual arithmetic mean of

geopotential heights of 500 hPa from years 1986 – 2011) and the L05 system ($N = 90$, arithmetic mean for lower bound

predictability curves; geometric mean up to 1999 and arithmetic mean after 1999 for upper bound predictability curves). It is

also based on knowledge of the largest Lyapunov exponent ($\lambda = 0.35$ day$^{-1}$) and the limit value of the predictability curve ($E_\infty$

$= 8.2$) of the L05 system.

Lyapunov exponents of the ECMWF forecasting system were recalculated by Eq. (14). The average value over all error growth

models for upper bound predictability is in the range $\langle 0.33;\ 0.41 \rangle$ day$^{-1}$ (Fig. 5a) and RMSEs are mostly about 0.01 day$^{-1}$. For

lower bound predictability curves average value over all error growth models is in the range $\langle 0.32;\ 0.41 \rangle$ day$^{-1}$ (Fig. 5b).

RMSEs are in the range $\langle 0.01;\ 0.02 \rangle$ day$^{-1}$. The average value over upper and lower bound predictability curves is shown in

Fig. 6 and RMSEs are mostly about 0.01 d$^{-1}$. Values of Lyapunov exponent are generally closer to parameters $\alpha^{EFS}$ of $Km_\beta$

, $Tm_\beta$ and $Gm$ than to $\alpha^{EFS}$ of $Km$ , $Tm$ and $Lm$ (Fig. 6).

New limit values were calculated from the error growth models by Eq. (15). For upper bound predictability curves, the average value over all error growth models is in the range $\langle 96; \, 133 \rangle$ m (Fig. 7a) and RMSEs are mostly about 1 m. For lower bound

predictability curves average value over all error growth models is in the range $\langle 114; \, 134 \rangle$ m (Fig. 7b) and RMSEs are mostly 3 m.

The argument that favors limit values calculated by Eq. (15) (Fig. 7, black full curves) instead of limit values calculated by Eq. (7) (Fig. 7, black dashed curves) is based on the new definition of model error (Eq. (16)) which shows a decreasing trend with increasing years for predictability curves with limit values calculated by Eq. (15), and almost constant trend with

increasing time (slight decrease can be due to the error of approximations) for predictability curves with limit values calculated by Eq. (7), which is theoretically impossible (Fig. 9a). This new model error calculated as a difference of model error parameters between the upper (Fig. 8a) and lower (Fig. 8b) bound predictability curves well support model error parameters calculated for upper bound predictability curves that are used to represents the intrinsic upscale error growth and propagation from small scales (Zhang et al., 2019).

## Code and data availability

The ECMWF forecasting system dataset was obtained from the personal repository of Linus Magnusson (Magnusson, 2013). L05 system dataset, products from the ECMWF forecasting system dataset, codes, and figures were conducted in Wolfram Mathematica and they are permanently stored at http://www.doi.org/10.17605/OSF.IO/CEK32.

**Author contributions**

HB proposed the idea, carried out the experiments, and wrote the paper. AR and JM supervised the study and co-authored the paper.

**Competing interests**

The authors declare that they have no conflict of interest.

**Acknowledgements**

The authors are grateful to Linus Magnusson for offering Dataset (ECMWF forecasting system) from his personal repository.





**Financial support**

This study was supported by the Czech Science Foundation, through grant 19-16066S.

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



| | Types of mean | Types of predictability curve | | | | | |
| --- | --- | --- | --- | --- | --- | --- | --- |
| | | Upper bound (U) | | | Lower bound (L) | | |
| ECMWF forecasting system (EFS) | Arithmetic (A) | $E_{U(A)}^{EFS}(t)$ $\alpha_{U(A)}^{EFS}$ | $E_{\infty,U(A)}^{EFS}$ $\beta_{U(A)}^{EFS}$ | $E_{\lim,U(A)}^{EFS}$ $p_{U(A)}^{EFS}$ | $E_{L(A)}^{EFS}(t)$ $\alpha_{L(A)}^{EFS}$ | $E_{\infty,L(A)}^{EFS}$ $\beta_{L(A)}^{EFS}$ | $E_{\lim,L(A)}^{EFS}$ $p_{L(A)}^{EFS}$ |
| | Geometric (G) | $E_{U(G)}^{EFS}(t)$ $\alpha_{U(G)}^{EFS}$ | $E_{\infty,U(G)}^{EFS}$ $\beta_{U(G)}^{EFS}$ | $E_{\lim,U(G)}^{EFS}$ $p_{U(G)}^{EFS}$ | $E_{L(G)}^{EFS}(t)$ $\alpha_{L(G)}^{EFS}$ | $E_{\infty,L(G)}^{EFS}$ $\bar{\beta}_{L(G)}^{EFS}$ | $E_{\lim,L(G)}^{EFS}$ $p_{L(G)}^{EFS}$ |
| L05 system (L05) | Arithmetic (A) | $E_{U(A)}^{L05}(t)$ $\alpha_{U(A)}^{L05}$ | $E_{\infty,U(A)}^{L05}$ $\beta_{U(A)}^{L05}$ | $E_{\lim,U(A)}^{L05}$ $p_{U(A)}^{EFS}$ | $E_{U(A)}^{L05}(t)$ $\alpha_{U(A)}^{EFS}$ | $E_{\infty,L(A)}^{EFS}$ $\beta_{L(A)}^{EFS}$ | $E_{\lim,L(A)}^{EFS}$ $p_{L(A)}^{EFS}$ |
| | Geometric (G) | $E_{U(G)}^{L05}(t)$ $\alpha_{U(G)}^{L05}$ | $E_{\infty,U(G)}^{L05}$ $\beta_{U(G)}^{L05}$ | $E_{\lim,U(G)}^{EFS}$ $p_{U(G)}^{EFS}$ | $E_{U(G)}^{EFS}(t)$ $\alpha_{U(G)}^{EFS}$ | $E_{\infty,L(G)}^{EFS}$ $\bar{\beta}_{L(G)}^{EFS}$ | $E_{\lim,L(G)}^{EFS}$ $p_{L(G)}^{EFS}$ |

**Table 1: Description of symbols that indicate types of predictability curve, types of mean and systems for prediction error** $E$, **theoretically calculated limit error** $E_{\infty}$, **and parameters of error growth models** $\alpha$, $\beta$, $p$ **and** $E_{\lim}$.






| $N$ | $\lambda^{L05}$ | $E_{\infty,U}^{L05}$ | $E_{\infty,L}^{L05}$ |
|---|---|---|---|
| 30 | 0.70 | 8.5 | 8.3 |
| 60 | 0.29 | 8.0 | 8.1 |
| 90 | 0.35 | 8.2 | 8.2 |
| 120 | 0.32 | 8.2 | 8.2 |
| 150 | 0.34 | 8.2 | 8.2 |
| 360 | 0.34 | | |

**Table 2: Values of the global largest Lyapunov exponents $\lambda^{L05}$ and limit values of predictability curves $E_{\infty,U}^{L05}$ a $E_{\infty,L}^{L05}$ for displayed number of variables $N$ of the L05 system.**





| RMS value | $KH_{PP}^{D}$ (day$^{-1}$) | $KH_{PM}^{D}$ (day$^{-1}$) | $KH_{PP}^{KP}$ (day$^{-1}$) | $KH_{PM}^{KP}$ (day$^{-1}$) | $OH$ (day$^{-1}$) | $LH$ (day$^{-1}$) |
|---|---|---|---|---|---|---|
| $\bar{\alpha}_{U(A)}^{L05}$ | 0.45 | 0.36 | 0.46 | 0.34 | 0.31 | 0.24 |
| $\bar{\alpha}_{L(A)}^{L05}$ | 0.46 | 0.40 | 0.48 | 0.41 | 0.33 | 0.23 |
| $\bar{\alpha}_{U(G)}^{L05}$ | 0.41 | 0.39 | 0.41 | 0.39 | 0.39 | 0.19 |
| $\bar{\alpha}_{L(G)}^{L05}$ | 0.42 | 0.40 | 0.43 | 0.41 | 0.35 | 0.19 |
| $\bar{\alpha}_{U(A)}^{EFS}$ | 0.45 | 0.41 | 0.46 | 0.39 | 0.36 | 0.21 |
| $\bar{\alpha}_{L(A)}^{EFS}$ | 0.48 | 0.42 | 0.50 | 0.40 | 0.35 | 0.27 |
|  | (-) | (-) | (-) | (-) | (-) | (-) |
| $\bar{E}_{\lim,U(A)}^{L05}$ | 7.5 | 7.8 | 7.3 | 7.8 | 8.2 | 8.9 |
| $\bar{E}_{\lim,L(A)}^{L05}$ | 7.5 | 7.8 | 7.3 | 7.6 | 8.3 | 9.3 |
| $\bar{E}_{\lim,U(G)}^{L05}$ | 7.7 | 7.8 | 7.7 | 7.8 | 7.8 | 11.0 |
| $\bar{E}_{\lim,L(G)}^{L05}$ | 7.8 | 8.0 | 7.6 | 7.8 | 8.3 | 10.6 |
|  | (m) | (m) | (m) | (m) | (m) | (m) |
| $\bar{E}_{\lim,U(A)}^{EFS}$ | 108 | 110 | 106 | 111 | 115 | 138 |
| $\bar{E}_{\lim,L(A)}^{EFS}$ | 114 | 117 | 112 | 117 | 123 | 134 |

**Table 3: RMS values calculated over all used initial errors for the L05 system ($N = 90$) and over all years for the ECMWF forecasting system of parameters $\bar{\alpha}$, $\bar{E}_{\lim}$ (for description see Table 1).**





| RMS value | $KH_{PM}^{D}$ (day$^{-1}$) | $KH_{PM}^{KP}$ (day$^{-1}$) | RMS value | OH (-) | RMS value | $KH_{PM}^{D}$ (day$^{-1}$) | $KH_{PM}^{KP}$ (day$^{-1}$) | RMS value | OH (-) |
|---|---|---|---|---|---|---|---|---|---|
| $\overline{\beta}_{U(A)}^{L05}$ | 0.21 | 0.27 | $\overline{p}_{U(A)}^{L05}$ | 0.3 | $\overline{\beta}_{U(G)}^{L05}$ | 0.03 | 0.04 | $\overline{p}_{U(G)}^{L05}$ | 0.9 |
| $\overline{\beta}_{L(A)}^{L05}$ | 0.10 | 0.12 | $\overline{p}_{L(A)}^{L05}$ | 0.4 | $\overline{\beta}_{L(G)}^{L05}$ | 0.04 | 0.03 | $\overline{p}_{L(G)}^{L05}$ | 0.7 |
|  | (m/day) | (m/day) |  | (-) |  | (m/day) | (m/day) |  | (-) |
| $\overline{\beta}_{U(A)}^{EFS}$ | 0.97 | 1.82 | $\overline{p}_{U(A)}^{EFS}$ | 0.6 | $\overline{\beta}_{L(A)}^{EFS}$ | 2.14 | 2.83 | $\overline{p}_{L(A)}^{EFS}$ | 0.40 |

**Table 4: RMS values calculated over all used initial errors for the L05 system ( $N = 90$ ) and over all years for the ECMWF forecasting system of parameters $\overline{\beta}$ and $\overline{p}$ (for description see Table 1).**




**Figure 1.** Comparison of upper bound predictability curves $E_{norm,U}$ of the ECMWF forecasting system normalized by $E_{\infty,U}^{EFS}$ (Eq. (15)) (EFS; annual arithmetic means, representative samples from 1986–2011) and the L05 system normalized by $E_{\infty,U}^{L05}$ (Table 2) (L05; geometric means (1986–1999), arithmetic means (2000–2011)).





**Figure 2.** **Comparison of lower bound predictability curves $E_{norm,L}$ of the ECMWF forecasting system normalized by $E_{\infty,L}^{EFS}$ (Eq. (15)), (EFS; annual arithmetic means, representative samples from1986–2011) and the L05 system normalized by $E_{\infty,L}^{L05}$ (Table 2) (L05; arithmetic mean).**



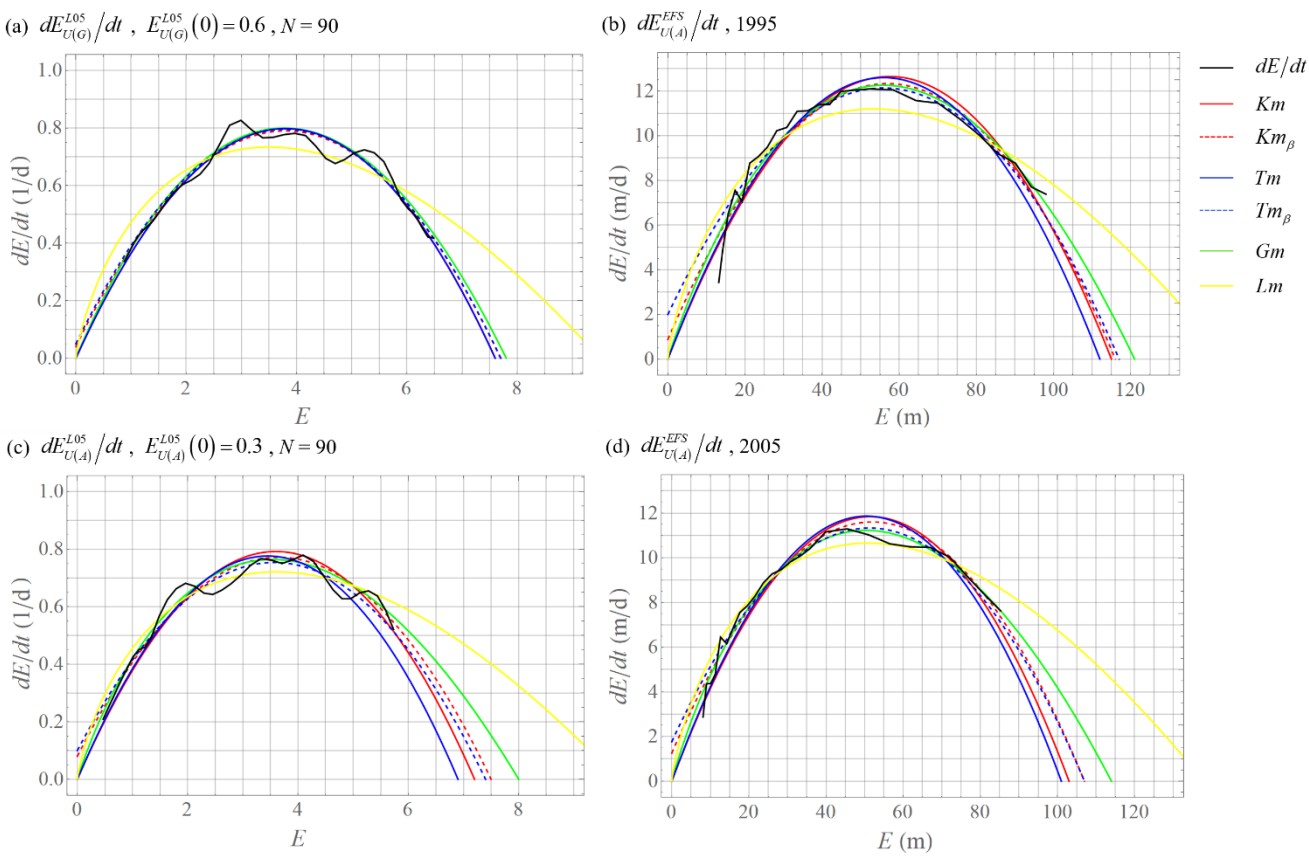

**Figure 3. Approximations of differences of upper bound predictability curves (representative samples). (a) – (b): the most similar**
**predictability curves in the year 1995 of the ECMWF forecasting system. (c) – (d): the most similar predictability curves in the year**
**2005 of the ECMWF forecasting system. Parameters from $Tm$ used in $Km$ (blue) and parameters from $Tm_\beta$ used in $Km_\beta$ (blue,**
**dashed).**





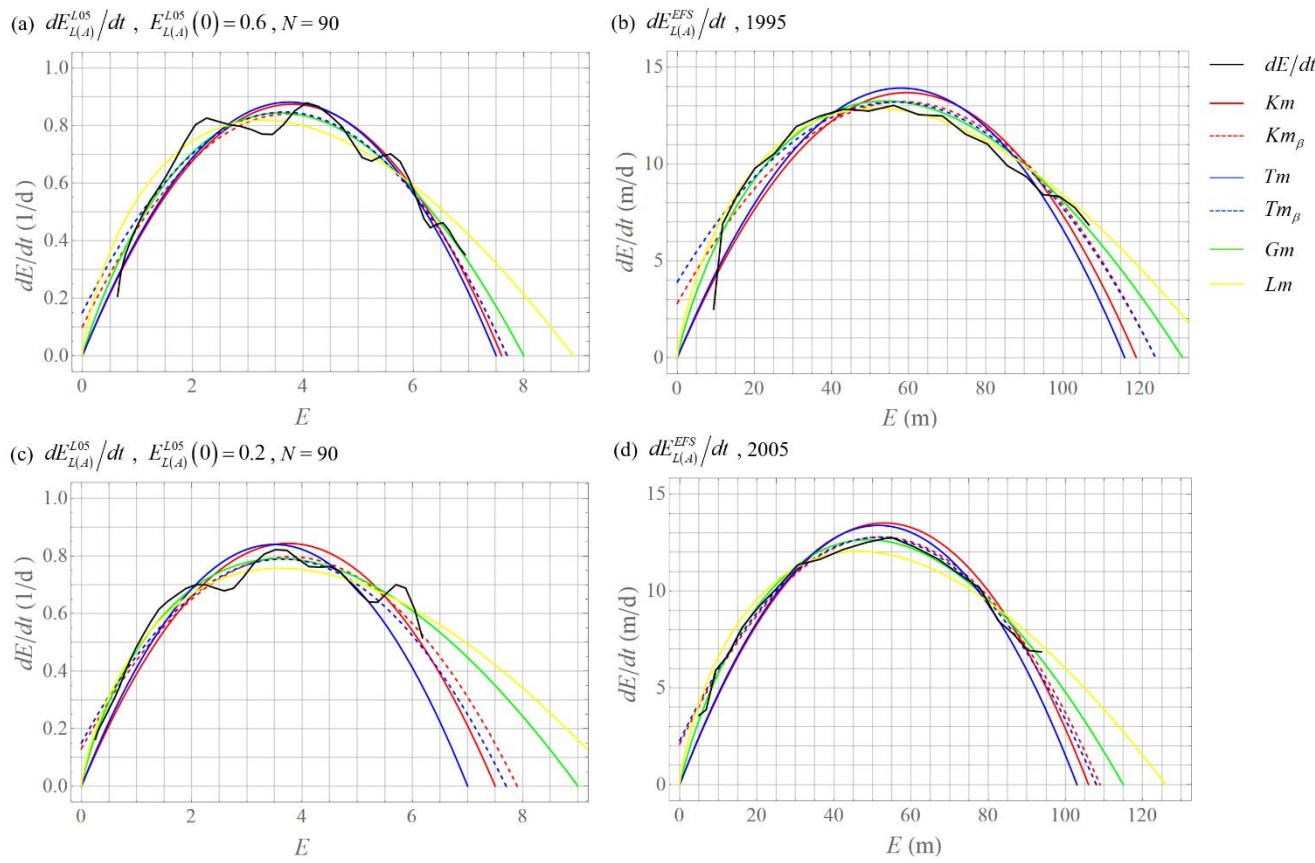

**Figure 4. Approximations of differences of lower bound predictability curves (representative samples). (a) – (b): the most similar**
**predictability curves in the year 1995 of the ECMWF forecasting system. (c) – (d): the most similar predictability curves in the year**
**2005 of the ECMWF forecasting system.** $Tm$ **displays parameters from** $Tm$ **used in** $Km$ **and** $Tm_\beta$ **displays parameters from** $Tm_\beta$

**used in** $Km_\beta$ **.**



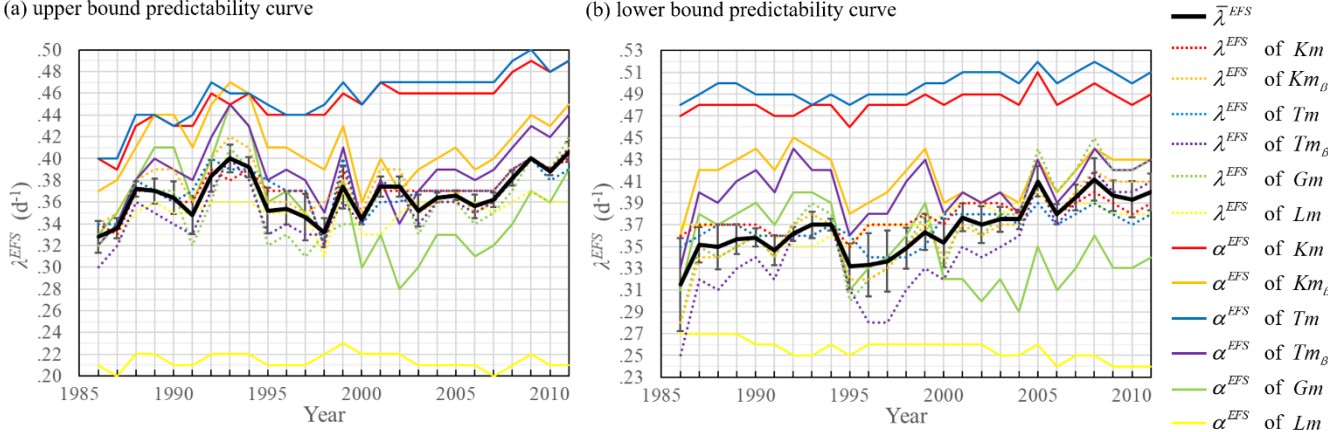

**Figure 5.** Lyapunov exponents $\lambda^{EFS}$ of the ECMWF forecasting system calculated by Eq. (14) and parameters $\alpha^{EFS}$ of error growth models for (a) upper and (b) lower bound predictability curves. $\overline{\lambda}^{EFS}$ is average value over all error growth models.



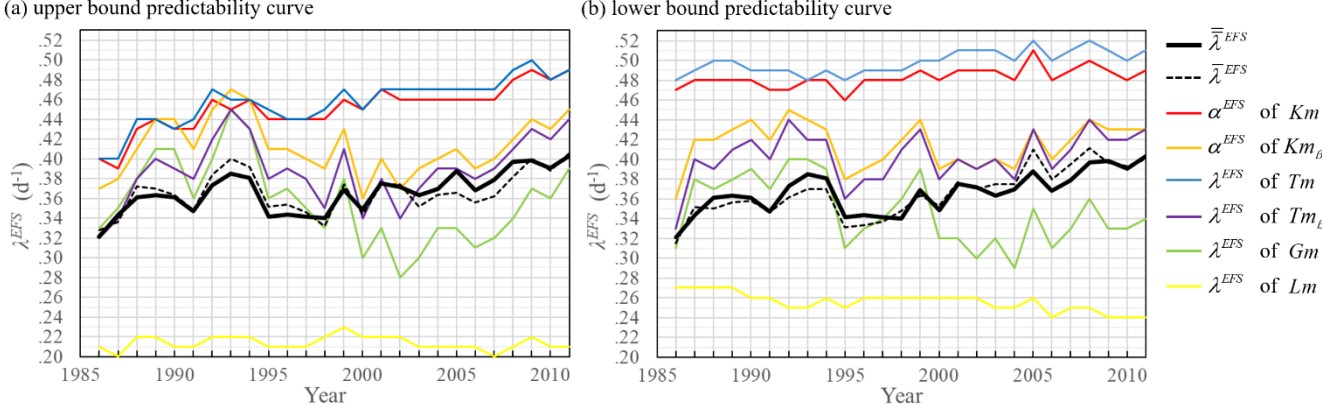

**Figure 6. Average values over upper and lower bound predictability curves of Lyapunov exponents** $\bar{\bar{\lambda}}^{EFS}$ **(black, solid), average values** $\bar{\lambda}^{EFS}$ **(black, dashed) for (a) upper and (b) lower bound predictability curves of the ECMWF forecasting system calculated by Eq. (14) and parameters** $\alpha^{EFS}$ **of error growth model for (a) upper and (b) lower bound predictability curves of the ECMWF forecasting system.**





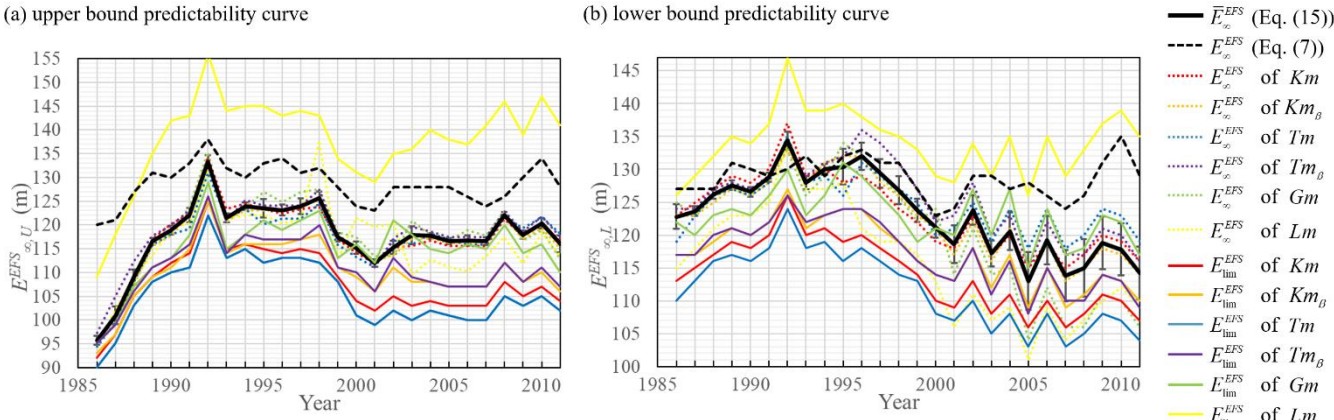

**Figure 7.** Limit values $E_\infty^{EFS}$ of the ECMWF forecasting system calculated by Eq. (15) and parameters $E_{\mathrm{lim}}^{EFS}$ of error growth models for (a) upper and (b) lower bound predictability curves. $\bar{E}_\infty^{EFS}$ (Eq. (15)) is average value over all error growth models and and $E_\infty^{EFS}$ (Eq. (7)) is limit values calculated by Eq. (7).






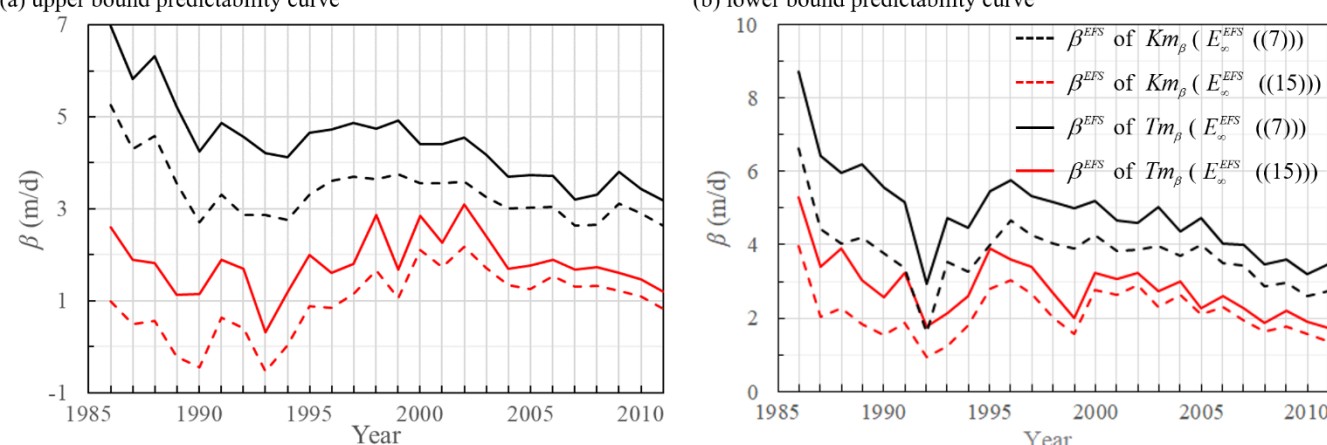

**Figure 8.** Parameters $\beta^{EFS}$ **(a) for upper bound predictability curves** $\beta_U^{EFS}$ **and (b) for lower bound predictability curves** $\beta_L^{EFS}$ **. Black curves represent** $\beta^{EFS}$ **approximated from predictability curves with** $E_\infty^{EFS}$ **calculated by Eq. (7), red curves pertain to** $\beta^{EFS}$ **approximated from predictability curves with** $E_\infty^{EFS}$ **calculated by Eq. (15), full curves correspond to** $\beta^{EFS}$ **calculated from** $Tm_\beta$ **and**

**dashed curves to** $\beta^{EFS}$ **calculated from** $Km_\beta$ **.**



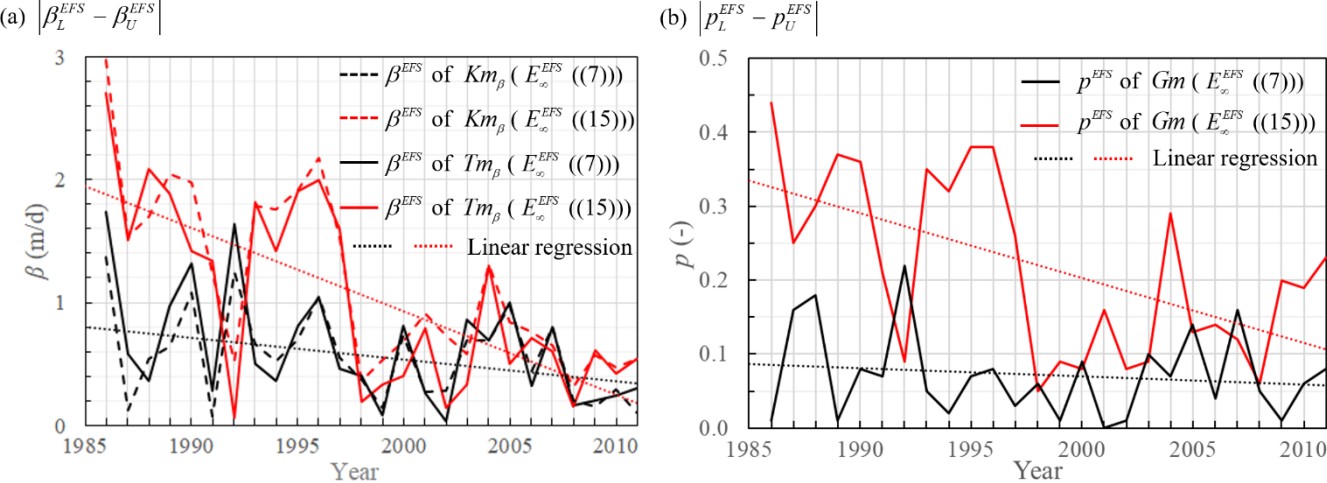

**Figure 9.** Absolute values of differences of parameters (a) $\left| \beta_L^{EFS} - \beta_U^{EFS} \right|$ and (b) $\left| p_L^{EFS} - p_U^{EFS} \right|$ between lower and upper bound predictability curves. For the notation see Fig. 8.