# Peer review of "Recalculation of error growth models' parameters for the ECMWF forecast system"

_Geoscientific Model Development, 2020_

## Referee Comment (RC1) · Anonymous Referee #1 · 17 Dec 2020

The reviewed manuscript aims to estimate the Lyapunov exponent, asymptotic error and model error for the ECMWF forecasts. The authors are comparing different parametric models to do such estimates. The topic is interesting but as a reader I have problems to follow some of the steps and assumptions in the manuscript, especially the correction step that is introduced. The manuscript also has a number of statements that need to be better clarified. I therefore recommend a major revision before it can be accepted for publication.

Major comments 1. I do not understand why the geometric vs arithmetic mean is discussed in the manuscript, especially as it cannot be fully applied to the ECMWF scores that are externally calculated. The part needs to be better motivated or removed.

2. In the L05 a model error is introduced, but it needs to be better explained how this

error would work and how it relates to real model errors.

3. The correction scheme for which the results are presented on line 210-233 is not properly introduced and motivated. For example, it is not easy to see how a correction based on L05 can be applied to ECMWF data. A proper description is needed.

Minor comments: Line 18-19: Initial errors grow due to the chaotic nature of the system. Line 19-20: The growth can be considered exponential for short lead times before non-linear effects (saturation) starts to play a role. Note that for very short lead times the error growth could be faster either due to small-scale processes as discussed in Zhang et al., or due to decorrelation between analysis error and forecast errors. Line 22: "with increasing" -> "as function of" Line 27: L is often referred to as practical predictability Line 30: Historically U is referred to as the perfect model assumption Line 38: Based on time-derivatives of the error Line 44 and other places: Do not use ' (e.g don't) Line 48: The need for a multi-scale growth model can be elaborated a bit more on. Line 99: Is "real" referring to the forecast as opposed to observations? Line 116: "Ago" and "ahead" is confusing Line 177: ERA-Interim does also include errors, which might be correlated with the forecast initial conditions Line 175: Do you tune the L05 differently for different years of ECMWF data, to account for lower initial and lower model errors? Line 191: "RMS" - Root mean square? Line 199: How would the result look if you force beta to be zero? Line 210-233: This paragraph is very difficult to follow. Line 244: Odd statement. Line 288: p should be given by the system and be independent of the model error Line 270-273: I do not understand the statement "used in ECMWF forecasting system". Please give a reference.
* * *

---

## Referee Comment (RC2) · Anonymous Referee #2 · 16 Mar 2021

Review of "Recalculation of error growth models' parameters for the ECMWF forecast system" by Hynek et al

Summary: This paper seeks to provide a new estimate of parameters of error growth models in the ECMWF forecasting system. Using a new approach, the authors calculate the largest Lyapunov exponent and two types of predictability curves, as well as the Lorenz's (2005) system, found that the largest Lyapunov exponent range from 0.32 to 0.41 day-1 in the ECMWF forecasting system, similar to the value of 0.35 day-1 in the Lorenz's system. Several results in this study are interesting, some parts could benefit from clarifications and major revisions. Below are the detailed comments. General Comments: 1. I found the paper not easy to read and understand, and it is not

well organized. There are too many symbols and many words are abbreviated that make reader confuse. 2. I'd suggest to divide section 2"Experimental setting"suggest into"2.1 Experimental setting"and "2.2 Calculation of the predictability curves". 3. The error growth estimate consists of initial and model error is lower bound predictability curve and the upper bound predictability curve only contains initial error. Can you say more about the differences between the bound predictability curves and the limit error? 4. L85:Remove the comma ","". It can be changed to "A bounded dynamical system with a positive Lyapunov exponent is chaotic". 5. L95: How to determine the values of N "real" and N "observed"? 6. L125-130: My main issue with this manuscript is that I'm not convinced that the measure of limit error really works, mainly because of the ERA-Interim daily data including uncertainty. Also, given that the maximum forecast time for the ECMWF forecasting system is 10 days, the forecast error may not be reach to the saturated value or predictability limit. 7. L125-130: What is the physical meaning of the 'limit error' you derived? Dose the limit error means the error of saturated value of predictability limit? 8. The paper of RuiqiangDing., and Jianping, Li(2011) is listed in References, but it cannot be found in the manuscript. Please check it again.

---

## Author Comment (AC2) · 28 Mar 2021

Dear referee, thank you for your comments. We are attaching a point-by-point reply to the comments and text changes. Best regards. Hynek Bednar

Please also note the supplement to this comment:
https://gmd.copernicus.org/preprints/gmd-2020-250/gmd-2020-250-AC2-supplement.pdf

---

## Author Response (AR1)

**(1) Comments from referee**

**Anonymous Referee #1**

The reviewed manuscript aims to estimate the Lyapunov exponent, asymptotic error
and model error for the ECMWF forecasts. The authors are comparing different parametric
models to do such estimates. The topic is interesting but as a reader I have
problems to follow some of the steps and assumptions in the manuscript, especially
the correction step that is introduced. The manuscript also has a number of statements
that need to be better clarified. I therefore recommend a major revision before it
can be accepted for publication.

**Major comments 1.**
 I do not understand why the geometric vs arithmetic mean is discussed
in the manuscript, especially as it cannot be fully applied to the ECMWF scores
that are externally calculated. The part needs to be better motivated or removed.
*2.* In the L05 a model error is introduced, but it needs to be better explained how this error would
work and how it relates to real model errors.
*3.* The correction scheme for which the results are presented on line 210-233 is not
properly introduced and motivated. For example, it is not easy to see how a correction
based on L05 can be applied to ECMWF data. A proper description is needed.

**Minor comments:**
*Line 18-19*: Initial errors grow due to the chaotic nature of the system.
*Line 19-20*: The growth can be considered exponential for short lead times before nonlinear effects
(saturation) starts to play a role. Note that for very short lead times the error growth could be faster
either due to small-scale processes as discussed in Zhang et al., or due to decorrelation between
analysis error and forecast errors.
*Line 22*: "with increasing" -> "as function of"
*Line 27*: L is often referred to as practical predictability
*Line 30*: Historically U is referred to as the perfect model assumption
*Line 38*: Based on time-derivatives of the error Line 44 and other places: Do not use ' (e.g don't)
*Line 48*: The need for a multi-scale growth model can be elaborated a bit more on.
*Line 99*: Is "real" referring to the forecast as opposed to observations?
*Line 116*: "Ago" and "ahead" is confusing
*Line 177*: ERA-Interim does also include errors, which might be correlated with the forecast initial
conditions
*Line 175*: Do you tune the L05 differently for different years of ECMWF data, to account for lower initial
and lower model errors?
*Line 191*: "RMS" - Root mean square?
*Line 199*: How would the result look if you force beta to be zero? Line 210-233: This paragraph is very
difficult to follow.
*Line 244*: Odd statement.
*Line 288*: p should be given by the system and be independent of the model error
*Line 270-273*: I do not understand the statement "used in ECMWF forecasting system". Please give a
reference.

**(2) author's response**

Dear referee,
thank you for your comments. We would like to respond to them:

Major comments
*1. I do not understand why the geometric vs arithmetic mean is discussed in the manuscript, especially as it cannot be fully applied to the ECMWF scores that are externally calculated. The part needs to be better motivated or removed.*
We added a better motivation (Line 179-180):
==Calculating predictability curves by arithmetic and geometric mean, although it does not affect predictability curves of the ECMWF forecasting system, is mentioned because it affects the calculation of predictability curves of the L05 system and this then affects the comparison of predictability curves, which is important for recalculation of error growth models' parameters for the ECMWF forecast system.==

*2. In the L05 a model error is introduced, but it needs to be better explained how this error would work and how it relates to real model errors.*
We added a more detailed explanation (Line 103-108):
==This method was presented by Lorenz (2005). Although not only resolution but also physical parameterization affects the deficiencies of the ECMWF system which make it different from the real atmosphere, Buizza (2010) showed that a comparison of predictability curves of the ECMWF system calculated from differences of prediction and analysis and from two predictions of systems with different horizontal resolutions leads to the same overall conclusions. Despite the sub differences mentioned by Buizza (2010), this method is sufficient for comparing the L05 system and the ECMWF forecasting system.==

*3. The correction scheme for which the results are presented on line 210-233 is not properly introduced and motivated. For example, it is not easy to see how a correction based on L05 can be applied to ECMWF data. A proper description is needed.*
We added a description (Line 228-232, 245-251): Some symbols were incorrectly marked in Eq. (15) and in its description ( $E_\infty^{EFS} = E_{\lim}^{EFS} + \left( \cancel{E}_{\lim}^{EFS} \cdot \left( E_\infty^{L05} - E_{\lim}^{L05} \right) \right) \Big/ E_\infty^{L05} \rightarrow E_\infty^{EFS} = E_{\lim}^{EFS} + \left( E_\infty^{EFS} \cdot \left( E_\infty^{L05} - E_{\lim}^{L05} \right) \right) \Big/ E_\infty^{L05}$ in Eq.,

$\cancel{E}_\infty^{L05} \rightarrow E_{\lim}^{L05}$ in description):
Line 228-232
==The formula (14) is based on the assumption, that if normalized predictability curves of the L05 system and the ECMWF forecasting system are similar, then the differences between true values of the global largest Lyapunov exponents ( $\lambda^{EFS}$ , $\lambda^{L05}$ ) and values determined from error growth models ( $\alpha^{EFS}$ , $\alpha^{L05}$ ) are similar ( $\lambda^{EFS} - \alpha^{EFS} \approx \lambda^{L05} - \alpha^{L05}$ ). Similarity of differences $\lambda - \alpha$ allows to estimate the global largest Lyapunov exponents of the ECMWF forecasting system.==

Line 245-251

$$E_\infty^{EFS} = E_{\lim}^{EFS} \cdot \frac{E_\infty^{L05}}{E_{\lim}^{L05}}, \tag{1}$$

where ==$E_{\lim}^{EFS}$== and $E_{\lim}^{L05}$ are values from error growth models and ==$E_\infty^{L05} = 8.2$ . As in calculating $\lambda^{EFS}$ , Eq. (15) based on the assumption, that if normalized predictability curves of the L05 system and the ECMWF forecasting==

==highlighted text==

system are similar, then the differences between true limit values ( $E_\infty^{EFS}$ , $E_\infty^{L05}$ ) and values determined from error growth models ( $E_{\lim}^{EFS}$ , $E_{\lim}^{L05}$ ) are similar. In this case, however, only normalized values can be compared:

$$\left(E_\infty^{EFS} - E_{\lim}^{EFS}\right)\Big/E_\infty^{EFS} \approx \left(E_\infty^{L05} - E_{\lim}^{L05}\right)\Big/E_\infty^{L05} \rightarrow E_\infty^{EFS} \approx E_{\lim}^{EFS} + \left(E_\infty^{EFS} \cdot \left(E_\infty^{L05} - E_{\lim}^{L05}\right)\right)\Big/E_\infty^{L05} \rightarrow E_\infty^{EFS} \approx E_{\lim}^{EFS} \cdot E_\infty^{L05}\Big/E_{\lim}^{L05}.$$

Similarity of normalized differences ( $\left(E_\infty - E_{\lim}\right)\Big/E_\infty$ ) allows to estimate new limit values of the ECMWF forecasting system.

Minor comments
*Line 18-19: Initial errors grow due to the chaotic nature of the system.*
Corrected (Line 19).

*Line 19-20: The growth can be considered exponential for short lead times before nonlinear effects (saturation) starts to play a role. Note that for very short lead times the error growth could be faster either due to small-scale processes as discussed in Zhang et al., or due to decorrelation between analysis error and forecast errors.*
Added (Line 22-23).

*Line 22: "with increasing" -> "as function of".*
Corrected (Line 23-24).

*Line 27: L is often referred to as practical predictability.*
Added (Line 28-29)

*Line 30: Historically U is referred to as the perfect model assumption.*
Added (Line 32)

*Line 38: Based on time-derivatives of the error.*
Corrected (Line 41-42).

*Line 44 and other places: Do not use ' (e.g don't).*
Corrected (Line 48, 49, 139, 162, 175).

*Line 48: The need for a multi-scale growth model can be elaborated a bit more on.*
Added (Line 52-53)
, where a parameter that usually measure model error, here represents the intrinsic upscale error growth and propagation from small scales.

*Line 99: Is "real" referring to the forecast as opposed to observations?*
Corrected (Line 109).
 predicted

*Line 116: "Ago" and "ahead" is confusing.*
Corrected (Line 127).
 ago

*Line 177: ERA-Interim does also include errors, which might be correlated with the forecast initial conditions.*
We did not find how this comment is associated with line 177.

*Line 175: Do you tune the L05 differently for different years of ECMWF data, to account for lower initial and lower model errors?*
Added (Line 163-165, 168-169, 186-187, 189-190).

Line 163-165:

these values are in the interval $E_U^{L05}(0) \in \langle 0.3; 0.8 \rangle$ , where lower values correspond to initial prediction errors of the ECMWF system from later years and higher values pertain to early years.

Line 168-169:

where lower values correspond to initial prediction errors of the ECMWF system from later years and higher values pertain to early years.

Line 186-187:

(for lower bound predictability curves this sets different values of the model error)

Line 189-190:

(the fact that this would mean unrealistic values of the model error for the ECMWF forecasting system is further discussed)

*Line 199: How would the result look if you force beta to be zero?*
Results with beta equal to zero are results of Quadratic ( $Km$ ) and hyperbolic tangent ( $Tm$ ) error growth models.

*Line 210-233: This paragraph is very difficult to follow.*
See Major comments *3*.

*Line 244: Odd statement.*
Deleted (Line 271)

*Line 288: p should be given by the system and be independent of the model error*
The answer can be found on the lines 275-278.

" $Gm$ has parameter $p$ that defines skewness of the originally parabolic shape of the difference of predictability curves. $p = 1$ pertains to symmetrical parabolic shape ( $Gm$ becomes $Km$ ) and $p = 0$ means the greatest skewness to the left ( $Gm$ becomes $Lm$ ). Parameters $\beta$ also skew the originally parabolic shape (Figs. 3 and 4). The model error can be seen as a difference between skewness of upper and lower bound predictability curves."

*Line 270-273: I do not understand the statement "used in ECMWF forecasting system". Please give a reference.*
Reference to Zhang et al. (2019) (Line 297).

**(3) authors changes in manuscript**

[revised manuscript text omitted]
. Using a new approach, the authors calculate the largest Lyapunov exponent and two types of predictability curves, as well as the Lorenz's (2005) system, found that the largest Lyapunov exponent range from 0.32 to 0.41 day-1 in the ECMWF forecasting system, similar to the value of 0.35 day-1 in the Lorenz's system. Several results in this study are interesting, some parts could benefit from clarifications and major revisions. Below are the detailed comments.

*General Comments*:
*1*. I found the paper not easy to read and understand, and it is not well organized. There are too many symbols and many words are abbreviated that make reader confuse.
*2*. I'd suggest to divide section 2"Experimental setting"suggest into"2.1 Experimental setting"and "2.2 Calculation of the predictability curves".
*3*. The error growth estimate consists of initial and model error is lower bound predictability curve and the upper bound predictability curve only contains initial error. Can you say more about the differences between the bound predictability curves and the limit error?
*4*. L85:Remove the comma ",,". It can be changed to "A bounded dynamical systém with a positive Lyapunov exponent is chaotic".
*5*. L95: How to determine the values of N "real" and N "observed"?
*6*. L125-130: My main issue with this manuscript is that I'm not convinced that the measure of limit error really works, mainly because of the ERAInterim daily data including uncertainty. Also, given that the maximum forecast time for the ECMWF forecasting system is 10 days, the forecast error may not be reach to the saturated value or predictability limit.
*7*. L125-130: What is the physical meaning of the 'limit error' you derived? Dose the limit error means the error of saturated value of predictability limit?
*8*. The paper of RuiqiangDing., and Jianping, Li(2011) is listed in References, but it cannot be found in the manuscript. Please check it again.

**(2) author's response**

Dear referee,

thank you for your comments. We would like to respond to them:

*1. I found the paper not easy to read and understand, and it is not well organized. There are too many symbols and many words are abbreviated that make reader confuse.*
To make it easier to read, we tried to create Table 1. (line 370), which we corrected, and at the same time, we added links to this table in the article.
Line 150: Description of symbols that indicate the type of prediction error $E$ in the text is provided in Table 1.
Line 214-215: Note that the description of symbols that indicate the type of parameters of error growth models $\alpha$ , $\beta$ , $p$ and $E_{lim}$ in the text is provided in Table 1.

*2. I'd suggest to divide section 2 "Experimental setting" suggest into "2.1 Experimental setting" and "2.2 Calculation of the predictability curves".*
We added:
Line 61: 2. Experimental setting
Line 91: 3. Calculation of the predictability curves
and we have modified the last paragraph of the introduction accordingly (Lines 57 - 59)

*3. The error growth estimate consists of initial and model error is lower bound predictability curve and the upper bound predictability curve only contains initial error. Can you say more about the differences between the bound predictability curves and the limit error?*
We have expanded the introductory definition:
Line 18 - 22: Forecast errors in numerical weather prediction systems grow in time because of the inaccuracy of the initial state (initial error), chaotic nature of the weather system itself, and the model imperfections (model error). The growth of forecast error in weather prediction is exponential on average. As an error becomes larger, its growth slows down and then stops with the magnitude saturating at about the average distance between two states chosen randomly from dynamically and statistically possible states (limit (saturated) error).

We also added information about the difference between the limit values of the lower and upper bound predictability curves:
Line 135 -136: $E_{\infty,U}$ and $E_{\infty,L}$ differ if the ECMWF forecasting system does not sufficiently describe the variability of the atmosphere (model error).

and we added a link to Figs. 3 and 4 to visually show the limit value as $dE/dt = 0$.
Line 194: $E_{lim}$ is the limit (saturated) value of $E$ (value of $E$ when $dE/dt = 0$, theoretically $E_\infty$ , Figs. 3 and 4)

*4. L85:Remove the comma ""''. It can be changed to "A bounded dynamical system with a positive Lyapunov exponent is chaotic".*
Line 85:The comma is removed.

*5. L95: How to determine the values of N "real" and N "observed"?*
We added the number of variables we tested.
Line 98:  ($N = 30$; $60$; $90$; $120$; $150$)
The reason why $N = 90$ was chosen is explained on lines 172-178.

*6. L125-130: My main issue with this manuscript is that I'm not convinced that the measure of limit error really works, mainly because of the ERAInterim daily data including uncertainty. Also, given that the maximum forecast time for the ECMWF forecasting system is 10 days, the forecast error may not be reach to the saturated value or predictability limit.*
We agree on this point. That is why we developed the method (Eq. (15)) that is independent of the calculation presented on the lines 125 – 130 (Eq. (7)), and we have shown that the method we specify is valid (5. Discussion)

*7. L125-130: What is the physical meaning of the 'limit error' you derived? Dose the limit error means the error of saturated value of predictability limit?*

In our text, the limit and the saturated value of the error have the same meaning.

Line 19 -22: The growth of forecast error in weather prediction is exponential on average. As an error becomes larger, its growth slows down and then stops with the magnitude saturating at about the average distance between two states chosen randomly from dynamically and statistically possible states (limit (saturated) error).

*8. The paper of RuiqiangDing., and Jianping, Li(2011) is listed in References, but it cannot be found in the manuscript. Please check it again.*

Thank you for your comment.

Line 164: which agrees with Ruiqiang and Jianping (2011)

(3) authors changes in manuscript

[revised manuscript text omitted]